# Dietary Patterns and Cardiometabolic Outcomes in Diabetes: A Summary of Systematic Reviews and Meta-Analyses

**DOI:** 10.3390/nu11092209

**Published:** 2019-09-13

**Authors:** Hana Kahleova, Jordi Salas-Salvadó, Dario Rahelić, Cyril WC Kendall, Emilie Rembert, John L Sievenpiper

**Affiliations:** 1Physicians Committee for Responsible Medicine, Washington, DC 20016, USA; hana.kahleova@gmail.com (H.K.); erembert@pcrm.org (E.R.); 2Institute for Clinical and Experimental Medicine, 140 21 Prague, Czech Republic; 3CIBER Fisiopatología de la Obesidad y Nutrición (CIBER Obn), Instituto de Salud Carlos III, 28029 Madrid, Spain; jordi.salas@urv.cat; 4Human Nutrition Unit, Biochemistry and Biotechnology Department, Sant Joan de Reus University Hospital, IISPV, Universitat Rovira i Virgili, 43201 Reus, Spain; 5Vuk Vrhovac University Clinic for Diabetes, Endocrinology and Metabolic Diseases, Merkur University Hospital, Zagreb 10000, Croatia; dario.rahelic@gmail.com; 6University of Zagreb School of Medicine, Zagreb 10000, Croatia; 7Department of Nutritional Sciences, Faculty of Medicine, University of Toronto, Toronto, ON M5S 1A1, Canada; cyril.kendall@utoronto.ca; 8Clinical Nutrition and Risk Factor Modification Center, St. Michael’s Hospital, Toronto, ON M5B 1W8, Canada; 9Toronto 3D Knowledge Synthesis and Clinical Trials Unit, St. Michael’s Hospital, Toronto, ON M5B 1W8, Canada; 10Division of Nutrition and Dietetics, College of Pharmacy and Nutrition, University of Saskatchewan, Saskatoon, SK S7N 5B5, Canada; 11Li Ka Shing Knowledge Institute, St. Michael’s Hospital, Toronto, ON M5B 1T8, Canada; 12Division of Endocrinology and Metabolism, St. Michael’s Hospital, Toronto, ON M5B 1W8, Canada

**Keywords:** cardiovascular disease, cardiometabolic outcomes, DASH, diabetes, dietary patterns, liquid meal replacements, Mediterranean, Nordic, portfolio, vegetarian

## Abstract

The Diabetes and Nutrition Study Group (DNSG) of the European Association for the Study of Diabetes (EASD) conducted a review of existing systematic reviews and meta-analyses to explain the relationship between different dietary patterns and patient-important cardiometabolic outcomes. To update the clinical practice guidelines for nutrition therapy in the prevention and management of diabetes, we summarize the evidence from these evidence syntheses for the Mediterranean, Dietary Approaches to Stop Hypertension (DASH), Portfolio, Nordic, liquid meal replacement, and vegetarian dietary patterns. The Grading of Recommendations Assessment, Development, and Evaluation (GRADE) approach was used to assess the quality of evidence. We summarized the evidence for disease incidence outcomes and risk factor outcomes using risk ratios (RRs) and mean differences (MDs) with 95% confidence intervals (CIs), respectively. The Mediterranean diet showed a cardiovascular disease (CVD) incidence (RR: 0.62; 95%CI, 0.50, 0.78), and non-significant CVD mortality (RR: 0.67; 95%CI, 0.45, 1.00) benefit. The DASH dietary pattern improved cardiometabolic risk factors (*P* < 0.05) and was associated with the decreased incidence of CVD (RR, 0.80; 95%CI, 0.76, 0.85). Vegetarian dietary patterns were associated with improved cardiometabolic risk factors (*P* < 0.05) and the reduced incidence (0.72; 95%CI: 0.61, 0.85) and mortality (RR, 0.78; 95%CI, 0.69, 0.88) of coronary heart disease. The Portfolio dietary pattern improved cardiometabolic risk factors and reduced estimated 10-year coronary heart disease (CHD) risk by 13% (−1.34% (95%CI, −2.19 to −0.49)). The Nordic dietary pattern was correlated with decreased CVD (0.93 (95%CI, 0.88, 0.99)) and stroke incidence (0.87 (95%CI, 0.77, 0.97)) and, along with liquid meal replacements, improved cardiometabolic risk factors (*P* < 0.05). The evidence was assessed as low to moderate certainty for most dietary patterns and outcome pairs. Current evidence suggests that the Mediterranean, DASH, Portfolio, Nordic, liquid meal replacement and vegetarian dietary patterns have cardiometabolic advantages in populations inclusive of diabetes.

## 1. Introduction

Diet-related cardiometabolic diseases, such as obesity, diabetes, and cardiovascular disease, inflict considerable implications on our health and economy [1]. Worldwide, the cumulative economic loss from chronic diseases between 2011 and 2030 is estimated at $17.3 trillion from healthcare costs, reduced productivity, and lost capital [2]. The global incidence of chronic disease and premature death is in large part due to suboptimal nutrition [3,4]. In fact, 45.5% of cardio-metabolic deaths in the United States have been associated with dietary habits, such as low fruit and vegetable consumption, and high intakes of sodium and processed meat [5]. A systematic analysis for the Global Burden of Disease Study 2017 assessed the health effects of dietary risks in 195 countries in 1990–2017 and estimated that 11 million deaths and 255 million disability-adjusted life-years were attributable to dietary risk factors, particularly high intake of sodium and low intakes of whole grains and fruits [6].

Simple pharmacotherapies typically reduce the risk of myocardial infarction by 20–30% [7], while healthy lifestyle choices may reduce it by up to 81–94% [8,9,10]. Interventions modifying intermediate risk factors, such as excess body weight, dyslipidemia, hypertension, prediabetes or diabetes, as well as improving lifestyle behaviors, i.e., avoiding tobacco, engaging in regular physical activity, and eating a balanced diet, are essential to preventing and treating cardiovascular disease. Nutrition may even surpass other habits, such as physical activity and no smoking, in preventing premature cardiovascular disease death and disability [3].

The relative risk for cardiovascular morbidity and mortality in adults with diabetes is approximately 2.5 to 5 times higher, compared with those without diabetes [11,12]. Therefore, up-to-date dietary recommendations, particularly for people with diabetes, are advisable.

To update the clinical practice guidelines for nutrition therapy in the prevention and management of diabetes, the Diabetes and Nutrition Study Group (DNSG) of the European Association for the Study of Diabetes (EASD) commissioned a series of systematic reviews and meta-analyses using the Grading and Recommendations Assessment, Development and Evaluation (GRADE) approach. This comprehensive overview of prospective cohort studies and randomized clinical trials pertains to different dietary patterns and patient-important cardiometabolic outcomes in populations including people with diabetes.

## 2. Materials and Methods

### 2.1. Design

Our review was conducted following the approach from the Cochrane Handbook for Systematic Reviews and Interventions [13] with reporting following the Meta-analysis of Observational Studies in Epidemiology (MOOSE) guideline [14] and Preferred Reporting Items for Systematic Reviews and Meta-Analyses (PRISMA) guideline [15]. Each review protocol was registered at ClinicalTrials.gov.

### 2.2. Data Sources and Searches

The databases MEDLINE (Medical Literature Analysis and Retrieval System Online), EMBASE (Excerpta Medica database), and the Cochrane Library were searched for relevant studies with restrictions for randomized clinical trials and prospective cohort studies for each dietary pattern and its association with cardiovascular outcomes and cardiometabolic risk factors, inclusive of people with diabetes. The complete search strategy is presented in Figure 1.

### 2.3. Data Extraction

The articles were independently reviewed by two reviewers and relevant data were extracted for each dietary pattern. Outcomes were the incidence and/or mortality of cardiovascular outcomes as well as cardiometabolic risk factors. The evidence for disease incidence outcomes and risk factor outcomes is presented as risk ratios (RR) and mean differences (MDs) with 95% confidence intervals (CIs), respectively. GRADE was used to assess the certainty of evidence.

### 2.4. Assessment of Study Quality

The risk of bias in the randomized controlled trials was assessed using the Cochrane Risk of Bias tool [16], which rates studies as having “high”, “low”, or “unclear” risk of bias across 6 domains: random sequence generation and allocation concealment (both selection bias), blinding of participants/study personnel (performance bias), blinding of outcome ascertainment (detection bias), incomplete outcome data (attrition bias), and selective reporting (reporting bias). The risk of bias in the observational studies was assessed using the Newcastle-Ottawa Scale (NOS). This scale, which awards up to nine points, awards for the cohort selection (max four points), comparability of cohort (max two points) and adequacy of the outcome measures (max three points) [17]. Studies that are awarded ≥6 points are considered “high” quality. Differences were reconciled by consensus.

### 2.5. Outcomes

The prespecified cardiometabolic outcomes included total cardiovascular mortality, coronary heart disease (CHD) mortality, stroke mortality, myocardial infarction mortality, cardiovascular disease (CVD) incidence, coronary heart disease (CHD) incidence, stroke incidence, myocardial infarction incidence, and diabetes incidence. Additionally, the outcomes included several cardiometabolic risk factors: hemoglobin A1c (HbA1c), fasting plasma glucose, fasting insulin, blood lipids (total cholesterol, low-density lipoprotein (LDL) cholesterol, high-density lipoprotein (HDL) cholesterol, non-HDL cholesterol, triglycerides, and apolipoprotein B (apo B)), body weight, body mass index (BMI), waist circumference, systolic and diastolic blood pressure, and C-reactive protein (CRP).

### 2.6. Statistical Analyses

Review Manager (RevMan), version 5.3 (Copenhagen, Denmark), was used to conduct primary and sensitivity analyses. STATA software, version 13.0 (College Station, TX, USA), was used to conduct subgroup and publication bias analyses. Using the generic inverse variance method, we pooled mean differences (MDs) for the trials and risk ratios (RRs) for the cohorts. Random-effects models, which yield more conservative summary effect estimates in the presence of residual heterogeneity, were used. Fixed-effects models were only used there were <5 included studies. Paired analyses were applied for crossover trials. Heterogeneity was assessed by the Cochran Q statistic and quantified by the I^2^ statistic. An I^2^ ≥ 50% and *P* < 0.10 was considered to be evidence of substantial heterogeneity. To explore potential causes of heterogeneity, we conducted sensitivity and a priori subgroup analyses. To perform the sensitivity analyses, we recalculated the pooled effect estimates and heterogeneity after removing each individual trial to determine whether a single study exerted an undue influence. A study whose removal changed the direction, significance, or magnitude (>10%) of the effect or the evidence of heterogeneity was considered influential. If ≥10 studies were available, then we explored sources of heterogeneity by prespecified subgroup analyses that included underlying disease status (diabetes, non-diabetes, etc.), diabetes duration, baseline values, study design (parallel, crossover), follow-up (<12 weeks, ≥12 weeks), variants of the dietary pattern (intervention), comparator, dose, risk of bias, and funding source (agency, industry, agency-industry, etc.) for the randomized controlled trials and age (children, adults), sex (female, male), variants of the dietary pattern (exposure), underlying disease status, follow-up (<10 years, ≥10 years), dose, validation of dietary assessment methods (validated, non-validated), NOS (<6, ≥6), and funding source for the prospective cohort studies. Meta-regression analyses assessed the significance of categorical and continuous subgroups analyses. Continuous meta-regression analyses and spline curve modelling (MKSPLINE procedure) were used to assess linear and non-linear dose–response analyses, respectively. If ≥10 studies were available, funnel plots and formal testing using the Egger and Begg tests were used to assess publication bias. If bias was detected, investigators used the Duval and Tweedie trim and fill method to input missing study data in attempt to adjust for funnel plot asymmetry.

### 2.7. Grading of the Evidence

The GRADE approach assessed the quality of the evidence [18,19,20,21,22,23,24,25,26,27,28,29] by grading the evidence as “high”, “moderate”, “low”, or “very low” certainty. Randomized controlled trial evidence begins as high certainty, and observational study (prospective cohort study) evidence begins as low certainty. Evidence can be downgraded due to study limitations (weight of studies showing risk of bias as assessed by the Cochrane Risk of Bias Tool [16] for randomized controlled trials or the Newcastle-Ottawa Scale [16] for observational studies), inconsistency (unexplained substantial heterogeneity, I^2^ ≥ 50% and *P* < 0.10), indirectness (presence of factors that limit the generalizability of the results in people with diabetes), imprecision (the 95% confidence intervals cross a minimally important difference), and publication bias (significant evidence of small-study effects). Evidence can be upgraded due to a large magnitude of association (relative risk, (RR) ≤0.5 or ≥2), a dose–response gradient, or attenuation by plausible confounding.

## 3. Results

### 3.1. Search Results

Figure 1 shows the process used to select the papers for the systematic reviews and meta-analyses (SRMAs) of the Mediterranean dietary pattern [30,31], vegetarian dietary pattern [32,33], liquid meal replacement dietary pattern [34], Nordic dietary pattern [35], Portfolio dietary pattern [36], and Dietary Approaches to Stop Hypertension (DASH) dietary pattern [37] reviewed in this paper.

### 3.2. Definitions of Dietary Patterns

The dietary patterns reviewed in this paper have broadly accepted definitions that likely vary between studies due to differences in populations. To best standardize dietary adherence, the studies used approaches, such as priori and posteriori scores and screener questions, based on the dietary pattern definitions listed below, or as specified otherwise.

The Mediterranean diet prioritizes vegetables, fruits, grains, legumes, nuts, virgin olive oil, and moderate amounts of fish and wine, over the consumption of red and processed meat, processed food and added sugars. The DASH diet focuses on vegetables, fruits, whole grains, legumes, fat-free or low-fat dairy, and nuts and limits the intake of cholesterol, total and saturated fat, red and processed meats, sweets and added sugars, including sugar-sweetened beverages. The Portfolio diet emphasizes four core cholesterol-lowering food components: nuts, plant protein from soy or other legumes, viscous soluble fiber, and plant sterols. The Nordic diet prioritizes vegetables, fruits, whole grains, legumes, rapeseed oil, fatty fish, shellfish, seaweed, and low-fat dairy. The vegetarian dietary pattern is based on a variety of fruits, vegetables, legumes and whole grains and excludes meat, poultry or fish. There are variations of the vegetarian diet, such as the lacto-ovo-vegetarian diet, which includes dairy and eggs, or a vegan diet, which excludes all animal products. Liquid meal replacements provide a mixture of macro- and micronutrients that are usually used to replace one or two main meals every day, and are often consumed with fruits, vegetables, and nuts.

### 3.3. Characteristics of the Review Papers

Full details are described in the published systematic reviews and meta-analyses.

The synthesis of the Mediterranean dietary pattern and cardiovascular disease and mortality in diabetes [30] reviewed 38 prospective cohort studies (*n* = 1,526,529) and three randomized clinical trials (*n* = 18,104). All studies and trials were inclusive of individuals with diabetes. Follow-up time ranged from 2 to 26 years in the cohort studies and 2 to 4.8 years in the randomized controlled trials (RCTs). The cohort studies were conducted in Asia, Europe, the USA, Australia, and internationally. The randomized controlled trials were conducted in India, Spain, and France.

The synthesis of the Mediterranean dietary pattern and cardiometabolic risk factors [31] summarized other meta-analyses, of which the characteristics were not detailed and can be found in the original papers.

The synthesis of the DASH dietary pattern and cardiometabolic outcomes [37] identified 15 prospective cohort studies (*n* = 942,140) and 33 controlled trials (*n* = 4479). All studies and trials were inclusive of individuals with diabetes. Follow-up time ranged from 5 to 24 years. The cohort studies were done in the USA, China, and Europe. The locations of the controlled trials were not mentioned.

The synthesis of the Portfolio dietary pattern and cardiometabolic outcomes [36] included seven controlled trial comparisons (*n* = 439). Individuals included in the trials had hyperlipidemia but were otherwise healthy and did not have diabetes. Follow-up time ranged from 4 to 24 weeks. All studies were carried out in Canada.

The synthesis of the Nordic dietary pattern and cardiometabolic outcomes is pending for publication [38]. Other systematic reviews and meta-analyses [35] of five randomized controlled trials (*n* = 513) have been published that indicate benefits of a Nordic diet. Individuals with and without diabetes were included. Follow-up time ranged from 2 weeks and 6 months. The trials were conducted in European countries. The intervention diet was either the Nordic diet according to the aforementioned description or a similar Nordic nutrition recommendation diet.

The synthesis of vegetarian dietary patterns and major cardiovascular outcomes [32] included seven prospective cohort studies (*n* = 197,737). Individuals with and without diabetes were included. Follow-up time varied between 5.5 and 21 years. Study locations were the USA, United Kingdom, and Germany. The studies compared non-vegetarians to different types of vegetarians, such as vegans (who exclude all meat, dairy, and eggs) and pescatarians (who exclude meat and poultry but eat fish).

The synthesis of vegetarian dietary patterns and cardiometabolic risk factors in diabetes [33] included nine randomized controlled trials (*n* = 664). Nearly all participants had diabetes (99%), for which most were taking medications (i.e., oral antihyperglycemic agents and insulin.) Follow-up time ranged from 4 to 74 weeks. Trials were conducted in the USA, Greece, Brazil, Czech Republic, and Korea. The vegetarian dietary patterns being observed included vegans and lacto-ovo-vegetarian (who exclude meats and fish but eat eggs and dairy).

The synthesis of liquid meal replacements and cardiometabolic risk factors [34] was of nine RCTs (*n* = 961). All participants had type 2 diabetes with HbA1c ranging from 6.5 to 8.8. Mean follow-up time ranged from 12–52 weeks. Trials were located in Asia, North America, Europe, and Australia. In order for a study to be included, the study intervention had to replace one to three meals with a liquid meal replacement.

### 3.4. Dietary Pattern and Cardiovascular Disease Mortality and Incidence

The meta-analysis of randomized clinical trials revealed a beneficial effect of the Mediterranean diet (Figure 2A) on total CVD incidence (RR: 0.62; 95%CI: 0.50, 0.78 with evidence of substantial inter-study heterogeneity: I^2^ = 86%; *P* = 0.01), total myocardial infarction incidence (RR: 0.65; 95%CI: 0.49, 0.88; with no evidence of inter-study heterogeneity: I^2^ = 50%; *P* = 0.16), coronary heart disease incidence (RR: 0.48; 95%CI: 0.33, 0.71), coronary heart disease mortality (RR: 0.33; 95%CI: 0.13, 0.85), stroke incidence (RR: 0.58; 95%CI: 0.42, 0.81), and a non-significant protective effect on cardiovascular disease mortality (RR: 0.67; 95%CI: 0.45, 1.00; with a high degree of inter-study heterogeneity: I^2^ = 64%; *P* = 0.09) and myocardial infarction mortality (RR: 0.67; 95%CI: 0.31, 1.43). The overall certainty of the evidence was moderate for total cardiovascular disease incidence, stroke incidence, and myocardial infarction incidence; low for total cardiovascular disease mortality, coronary heart disease incidence and mortality; and very low for myocardial infarction mortality.

The meta-analysis of prospective cohort studies in populations inclusive of individuals with diabetes (Figure 2B) compared the outcomes of individuals with the highest vs. individuals with the lowest adherence to the Mediterranean diet. An inverse association was found with total cardiovascular disease mortality (RR: 0.79; 95%CI: 0.77, 0.82; with no evidence of inter-study heterogeneity: I^2^ = 0%; *P* = 0.64), coronary heart disease incidence (RR: 0.73; 95%CI: 0.62, 0.86; with no evidence of inter-study heterogeneity: I^2^ = 26%; *P* = 0.23), coronary heart disease mortality (RR: 0.73; 95%CI: 0.59, 0.89; with evidence of substantial inter-study heterogeneity: I^2^ = 63%; *P* = 0.02), stroke incidence (RR: 0.80; 95%CI: 0.71, 0.90; with no evidence of inter-study heterogeneity: I^2^ = 0%; *P* = 0.63), stroke mortality (RR: 0.87; 95%CI: 0.80, 0.96; with no evidence of inter-study heterogeneity: I^2^ = 0%; *P* = 0.74), and myocardial infarction incidence (RR: 0.73; 95%CI: 0.61, 0.88; with no evidence of inter-study heterogeneity: I^2^ = 0%; *P* = 0.66). The association with total cardiovascular disease incidence was nonsignificant (RR: 0.88; 95%CI: 0.74, 1.03; with evidence of substantial inter-study heterogeneity: I^2^ = 53%; *P* = 0.04). The overall certainty of the evidence was moderate for total cardiovascular disease mortality and coronary heart disease incidence; low for coronary heart disease mortality and stroke incidence; and very low for total cardiovascular disease incidence, stroke mortality and myocardial infarction incidence.

The DASH dietary pattern (Figure 2C) was associated with reduced risk of cardiovascular disease (RR, 0.80; 95%CI: 0.76 to 0.85; with no evidence of inter-study heterogeneity: I^2^ = 30%; *P* = 0.16), as well as coronary heart disease (0.79; 95%CI: 0.71 to 0.88; with no evidence of inter-study heterogeneity: I^2^ = 0%; *P* = 0.58), and stroke (0.81; 95%CI: 0.72 to 0.92; with no evidence of inter-study heterogeneity: I^2^ = 0%; *P* = 0.91) in prospective cohort studies. The overall certainty of the evidence was low for total cardiovascular disease and stroke risk; and very low for coronary heart disease risk.

The meta-analysis of prospective cohort studies on the vegetarian dietary pattern (Figure 2D) adherence was associated with decreased risk of coronary heart disease mortality (RR, 0.78; 95%CI: 0.69, 0.88; with no evidence of inter-study heterogeneity: I^2^ = 46%; *P* = 0.07) and incidence (0.72; 95%CI: 0.61, 0.85). A weak non-significant association was found with cardiovascular disease mortality (0.92; 95%CI: 0.84, 1.02; with no evidence of inter-study heterogeneity: I^2^ = 34%; *P* = 0.18) and stroke mortality (0.92; 95%CI: 0.77, 1.10; with no evidence of inter-study heterogeneity: I^2^ = 44%; *P* = 0.13). Due to indirectness and imprecision, all outcomes were downgraded to “very low” certainty of evidence.

### 3.5. Dietary Pattern and Cardiometabolic Outcomes

The Mediterranean diet (Figure 3A) significantly decreased body weight (−1.75 kg (95%CI, −2.86 to −0.64); with evidence of substantial inter-study heterogeneity: I^2^ = 95%; *P* < 0.001), waist circumference (−0.54 cm (95%CI: −0.77, −0.31); with evidence of substantial inter-study heterogeneity: I^2^ = 96%; *P* < 0.001), fasting plasma glucose (−0.50 mmol/L (95%CI: −0.81, −0.20); with substantial inter-study heterogeneity: I^2^ = 97%), LDL cholesterol (−0.07 mmol/L (95%CI: −0.13, −0.01); with no evidence of inter-study heterogeneity: I^2^ = 22%; *P* = 0.27), triglycerides (−0.46 mmol/L (95%CI: −0.72, −0.21); with substantial inter-study heterogeneity: I^2^ = 94%; *P* < 0.001), systolic blood pressure (−0.72 mmHg (95%CI, −1.03 to −0.42); with evidence of substantial inter-study heterogeneity: I^2^ = 97%; *P* < 0.001), as well as diastolic blood pressure (−0.94 mmHg (95%CI, −1.45 to −0.44); with evidence of substantial inter-study heterogeneity: I^2^ = 99%; *P* < 0.001).

The DASH dietary pattern (Figure 3B) significantly decreased systolic blood pressure (−5.2 mmHg (95%CI, −7.0 to −3.4); with evidence of substantial inter-study heterogeneity: I^2^ = 76%; *P* < 0.001), as well as diastolic blood pressure (−2.60 mmHg (95%CI, −3.50 to −1.70); with evidence of inter-study heterogeneity: I^2^ = 49%; *P* = 0.009), total cholesterol (−7.9 mg/dL (95%CI, −12.00 to −3.80); with evidence of inter-study heterogeneity: I^2^ = 52%; *P* = 0.01), LDL cholesterol (95%CI, −4.00 mg/dL (−7.70 to −0.30); with no evidence of inter-study heterogeneity: I^2^ = 37%; *P* = 0.08), fasting plasma insulin (−0.15 uU/mL (95%CI, −0.22 to −0.08); with no evidence of inter-study heterogeneity: I^2^ = 0%; *P* = 0.49), HbA1c (−0.53% (95%CI, −0.62 to −0.43); with evidence of a high inter-study heterogeneity: I^2^ = 99%; *P* < 0.001), and body weight (−1.42 kg (95%CI, −2.03 to −0.82); with evidence of substantial inter-study heterogeneity: I^2^ = 71%; *P* < 0.001). There was no effect on HDL cholesterol, triglycerides, fasting plasma glucose, homeostatic model assessment of insulin resistance (HOMA-IR), or CRP. The overall certainty of the evidence was graded as moderate for systolic blood pressure, LDL cholesterol, fasting plasma insulin, HOMA-IR, and body weight; and low for CRP, total cholesterol, HDL cholesterol, diastolic blood pressure, triglycerides, fasting plasma glucose, and HbA1c.

The Portfolio dietary pattern (Figure 3C) significantly reduced LDL cholesterol by ~17% (MD = −0.73 mmol/L; (95%CI, −0.89, −0.56); with evidence of substantial inter-study heterogeneity: I^2^ = 67%; *P* = 0.006) as well as other lipid outcomes, such as total cholesterol (−0.81 mmol/L (95%CI, −0.98, −0.64); with evidence of low inter-study heterogeneity: I^2^ = 52%; *P* = 0.05), non-HDL cholesterol (−0.83 mmol/L (95%CI, −1.03, −0.64); with evidence of substantial inter-study heterogeneity: I^2^ = 61%; *P* = 0.02), triglycerides (−0.28 mmol/L (95%CI, −0.42, −0.14); with evidence of substantial inter-study heterogeneity: I^2^ = 58%; *P* = 0.03), and apolipoprotein B (−0.19 g/L (95%CI, −0.23, −0.15); with evidence of substantial inter-study heterogeneity: I^2^ = 60%; *P* = 0.02). Other cardiometabolic risk factors were also significantly reduced by the Portfolio dietary pattern, including systolic blood pressure (−1.75 mmHg (95%CI, −3.23, −0.26) with no evidence of inter-study heterogeneity: I^2^ = 0%; *P* = 0.79), diastolic blood pressure (−1.36 mmHg (95%CI, −2.33, −0.38) with no evidence of inter-study heterogeneity: I^2^ = 0%; *P* = 0.46), CRP (−0.58 mg/L (95%CI, −1.01, −0.15) with no evidence of inter-study heterogeneity: I^2^ = 33%; *P* = 0.18), and 10-year coronary heart disease risk estimated by Framingham risk score by 13% (−1.34% (95%CI, −2.19 to −0.49) with no evidence of inter-study heterogeneity: I^2^ = 54%; *P* = 0.07). HDL cholesterol and body weight were unaffected. The overall certainty of the evidence was high for LDL cholesterol, total cholesterol, non-HDL cholesterol, triglycerides, apolipoprotein B and body weight and moderate for HDL cholesterol, systolic blood pressure, diastolic blood pressure, CRP and estimated 10-year coronary heart disease risk.

Liquid meal replacements (Figure 3D) as part of a weight loss diet in individuals with diabetes significantly reduced body weight (−2.37 kg (95%CI, −3.30, −1.44); with evidence of high inter-study heterogeneity: I^2^ = 84%; *P* < 0.001), BMI (−0.87 kg/m^2^ (95%CI, −1.32, −0.42); with evidence of high inter-study heterogeneity: I^2^ = 89%; *P* < 0.001), body fat (−1.66% (95%CI, −2.17, −1.15); with no evidence of inter-study heterogeneity: I^2^ = 50%; *P* = 0.11), waist circumference (−2.24 cm (95%CI: −3.72, −0.77); with evidence of substantial inter-study heterogeneity: I^2^ = 74%; *P* =0.004), HbA1c (−0.43% (95%CI, −0.66, −0.19); with evidence of high inter-study heterogeneity: I^2^ = 87%; *P* < 0.001), fasting plasma glucose (−0.63 mmol/L (95%CI, −0.99, −0.27); with evidence of substantial inter-study heterogeneity: I^2^ = 70%; *P* < 0.001), fasting plasma insulin (−11.8 pmol/L (95%CI, −23.1, −0.54); with no evidence of inter-study heterogeneity: I^2^=22%; *P* = 0.27), systolic (−4.97 mmHg (95%CI, −7.32, −2.62); with evidence no evidence of inter-study heterogeneity: I^2^ = 53%; *P* = 0.05) and diastolic blood pressure (−1.98 mmHg (95%CI, −3.05, −0.91); with no evidence of inter-study heterogeneity: I^2^ = 15%; *P* = 0.32). There was no significant effect on LDL cholesterol, HDL cholesterol, non-HDL cholesterol and triglycerides (*P* > 0.05). The overall certainty of the evidence was graded as moderate for body weight, BMI and body fat; low for waist circumference; high for systolic blood pressure; moderate for fasting insulin, non-HDL cholesterol, and diastolic blood pressure; and low for HbA1c, fasting glucose, LDL-cholesterol, HDL-cholesterol, and triglycerides.

Our review on the Nordic diet, in line with the protocol outlined in the methods, is pending. Preliminary results of our systematic review and meta-analysis of the prospective cohort studies showed beneficial associations between the Nordic dietary pattern and reduced risk of CVD (0.93 (95%CI, 0.88, 0.99)) and stroke incidence (0.87 (95%CI, 0.77, 0.97)), but not mortality. All outcomes were graded very low for overall certainty of evidence [38]. The results from our systematic review and meta-analysis of randomized controlled trials are not yet available. A systematic review meta-analysis of randomized controlled trials was recently reported by Ramezani-Jolfaie et al. [35]. They (Figure not shown) showed that the Nordic dietary pattern lowered total cholesterol (−0.39 mmol/L (95%CI −0.76, −0.01) with evidence of high inter-study heterogeneity: I^2^ = 91.7%; *P* < 0.001) and LDL cholesterol (−0.30 mmol/L (95%CI −0.54, −0.06) with evidence of substantial inter-study heterogeneity: I^2^ = 87.8%; *P* < 0.001) compared with the control groups however. No significant changes were seen in HDL cholesterol and triglycerides. The Nordic dietary pattern significantly reduced the systolic (− 3.97 mmHg (95%CI, −6.40, −1.54) with no evidence of inter-study heterogeneity: I^2^ = 26.1%; *P* = 0.26) and diastolic blood pressure (−2.08 mmHg (95%CI, −3.44, −0.73) with no evidence of inter-study heterogeneity: I^2^ = 0%; *P* = 0.49). The Nordic dietary pattern has also reduced body weight, insulin resistance, and improved blood lipid profiles in randomized controlled trials of individuals with obesity or metabolic syndrome [35,39,40].

Vegetarian dietary patterns (Figure 3E) significantly lowered HbA_1c_ (MD = −0.29% (95%CI: −0.45, −0.12); with no evidence of inter-study heterogeneity: I^2^ = 14%; *P* = 0.32), fasting glucose (−0.56 mmol/L (95%CI: −0.99, −0.13); with no evidence of inter-study heterogeneity: I^2^ = 0%; *P* = 0.56), LDL cholesterol (−0.12 mmol/L (95%CI: −0.20, −0.04); with no evidence of inter-study heterogeneity: I^2^ = 0%; *P* = 0.54), non-HDL-C (−0.13 mmol/L (95%CI:−0.26, −0.01); with no evidence of inter-study heterogeneity: I^2^ = 0%; *P* = 0.44), body weight (−2.15 kg (95%CI: −2.95, −1.34); with no evidence of inter-study heterogeneity: I^2^ = 21%; *P* = 0.28), BMI (−0.74 kg/m^2^ (95%CI: −1.09, −0.39); with evidence of substantial inter-study heterogeneity: I^2^ = 60%; *P* = 0.03) and waist circumference (−2.86 cm (95%CI: −3.76, −1.96); with no evidence of inter-study heterogeneity: I^2^ = 48%; *P* = 0.12). No significant effects on HDL cholesterol, fasting insulin, triglycerides or blood pressure were observed. HbA1c, fasting plasma glucose, LDL cholesterol, HDL cholesterol, non-HDL cholesterol, systolic and diastolic blood pressure and body weight were all graded moderate for overall certainty of evidence; fasting insulin, triglycerides and waist circumference were graded low.

## 4. Discussion

### 4.1. Summary of Main Findings

Our findings demonstrate that the Mediterranean, DASH, Portfolio, and vegetarian dietary patterns play valuable roles in reducing the incidence and mortality from various cardiovascular disease outcomes in individuals with and without diabetes.

Our pooled analysis of randomized clinical trials shows that the Mediterranean diet is associated with a 38% lower risk of cardiovascular disease and a nonsignificant reduction in cardiovascular disease mortality. The pooled effect from the prospective cohort studies shows a reduction in cardiovascular disease incidence and mortality by 12% and 21%, respectively.

Our meta-analysis demonstrates a 20% reduced risk of cardiovascular disease from adherence to the DASH dietary pattern. This dietary pattern also shows significantly lower diastolic and systolic blood pressure, which may translate to an approximately 20% reduction in risk of cardiovascular disease, along with meaningful benefits in other established cardiovascular risk factors in those with and without diabetes.

The pooled analyses of prospective cohort studies demonstrate that vegetarian dietary patterns are associated with a 28% reduced risk of coronary heart disease and 22% reduced coronary heart disease mortality. Furthermore, the pooled analyses of randomized clinical trials have demonstrated clinically meaningful reductions in body weight, in LDL cholesterol and in HbA1c with the control diets.

In randomized clinical trials, the Mediterranean, diet decreased fasting plasma glucose, systolic blood pressure, diastolic blood pressure, body weight, waist circumference, LDL cholesterol and triglycerides. The DASH diet decreased fasting blood insulin, systolic blood pressure, diastolic blood pressure, body weight, total cholesterol and LDL cholesterol. The Portfolio diet led to clinically significant improvements in LDL cholesterol, other established cardiometabolic risk factors, and 10-year coronary heart disease risk. The Nordic diet reduced systolic blood pressure, diastolic blood pressure, LDL cholesterol, and total cholesterol. The vegetarian diet improved body weight/adiposity, LDL cholesterol, non-HDL cholesterol, and glycemic control in individuals with diabetes. Weight loss diets incorporating liquid meal replacements led to modest improvements in adiposity, glycemic control and blood pressure without harming blood lipids. These findings are similar to those observed under the Mediterranean, DASH, Portfolio, and vegetarian dietary patterns. The majority of the randomized controlled trials instructed the control participants to make no diet change. Others assigned the control group to specific diets, such as a low-fat diet, a typical weight loss diet, or a conventional diabetes diet.

### 4.2. Results in Relation to Other Studies

Our findings support those of preceding systematic reviews and meta-analyses. This paper presents the most up-to-date information available, utilizing more data than previous works, and using the GRADE approach to grade the evidence. The benefits of the Mediterranean diet on various cardiovascular outcomes have been reported previously [31]. Previous meta-analyses have shown a 41% lower risk of CVD mortality [41] and a 27% lower risk of CVD incidence [42] when comparing high versus low adherence to this dietary pattern.

Randomized controlled trials have shown that the DASH diet decreases LDL cholesterol, blood pressure, and other cardiometabolic risk factors. Prospective cohort studies have shown decreased diabetes and cardiovascular mortality in response to the DASH diet [43,44,45].

The Portfolio dietary pattern emphasizes four individual food components, each one of which appears to significantly decrease LDL cholesterol by: 7% for 2 g/day of plant sterols/stanols [46], 7% for 5–10 g/day of viscous fibers [47], 4–5% for ~30 g/day of plant proteins [48,49,50], and 7% for 67 g/day of nuts [51]. Within the Portfolio dietary pattern, the predicted additive effect of these four food components on LDL cholesterol reduction is ~21%.

The effects of the Nordic diet on cardiometabolic outcomes are less studied than other dietary patterns. From the limited evidence, it appears that the Nordic diet has beneficial effects on blood pressure, as previously described in a meta-analysis by Ndanuko et al. [52]. However, a meta-analysis of three cross-sectional studies has not shown any benefits of the Nordic diet on lipid profiles or blood pressure [53]. The Nordic diet appears to influence blood pressure more than the Mediterranean diet, but less than the DASH diet. A meta-analysis of eight cohort studies by Massara et al. found the Nordic diet to be protective against CVD and stroke incidence [38]. The meta-analysis did not show significant associations with CVD mortality and CHD incidence. While the current evidence is limited, the Nordic diet seems to have promising effects on cardiometabolic outcomes.

Vegetarian dietary patterns tend to contain more fiber, plant protein, antioxidants, and phytochemicals, and less saturated fat than non-vegetarian dietary patterns [54]. Current systematic reviews and meta-analyses have found that the consumption of a vegetarian dietary pattern is associated with lower risk of CHD in prospective cohort studies [55], and improved cardiometabolic risk factors in randomized controlled trials in those with and without diabetes, compared to non-vegetarian dietary patterns [33,56,57].

Currently, the Mediterranean, DASH, and vegetarian dietary patterns are all recommended as healthy by the *2015–2020 Dietary Guidelines for Americans* [58] and are included in the American Diabetes Association’s clinical practice guidelines for people with diabetes or prediabetes [59]. Additionally, the health benefits of dietary patterns reviewed in this paper are consistent with recent literature looking at the effects of food groups on various health outcomes. A 2019 review has shown an inverse association between the incidence of type 2 diabetes and increased intake of whole grains and cereal fiber, both of which are commonly recommended among the dietary patterns discussed in our paper [60]. The review also found a positive association between the incidence of type 2 diabetes and higher intake of red meat, processed meat, and sugar-sweetened beverages, all of which are commonly limited and/or advised against in the discussed dietary patterns. Additional literature supports greater intakes of high-fiber food groups, such as whole grains, nuts, fruits and vegetables to be associated with lower risk of cardiovascular disease, stroke, and type 2 diabetes [4,61,62].

### 4.3. Potential Mechanisms

The observed benefits of the Mediterranean, DASH, Portfolio, Nordic and vegetarian dietary patterns on cardiometabolic risk factors may be explained by different potential mechanisms. These dietary patterns emphasize eating plant foods, which are inherently high in fiber [60,61]. Fiber aids in weight loss and, in turn, improvements in blood sugar [62]. Additionally, dietary fiber increases satiety and thus reduces energy intake [63]. Furthermore, these plant foods are also typically lower on the glycemic index and in saturated fat, and higher in unsaturated fat, plant protein, and phytochemicals. Individually, all of these components have shown beneficial effects on various cardiometabolic risk factors, of which the respective mechanisms have been previously described [31,64,65,66,67,68].

All dietary patterns include the four individual food components that are specifically emphasized under the Portfolio dietary pattern: plant/sterols/stanols, viscous fibers, plant protein, and nuts. The beneficial effects of these components may be due to different mechanisms of action. For instance, plant sterols/stanols may inhibit the absorption of cholesterol in the small intestine [69]. Viscous fibers may increase the rate of bile acid excretion [70] and the production of short-chain fatty acids in the colon, and may reduce LDL cholesterol levels by affecting cholesterol synthesis [71]. Plant protein may act as a vehicle for the plant sterols/stanols and viscous fiber, or other antiatherogenic agents [72]. and certain amino acids common in plant proteins may decrease cholesterol levels [73]; The effects of nuts, such as almonds, likely stem from the nutrients they provide since they contain phytosterols, fiber, and plant proteins [74].

Although all these dietary patterns emphasize vegetables, fruits, legumes and whole grains, there are also some differences. Vegetarian diets exclude all meat and vegan diets do not contain any animal products. In contrast, the Mediterranean and the DASH diet encourage white meat and low-fat dairy consumption. Furthermore, while the DASH, Portfolio, the Nordic and vegetarian diets are typically fairly low in fat (especially saturated fat), the Mediterranean diet is characterized by a high fat content, coming particularly from monounsaturated and polyunsaturated fatty acids.

### 4.4. Strengths and Limitations

This series of systematic reviews and meta-analyses had several strengths. First of all, we have conducted an in-depth search and identified all available randomized controlled trials and prospective cohort studies examining the effect of the Mediterranean, the DASH, the Portfolio, and the vegetarian dietary patterns, as well as liquid meal replacements, on cardiometabolic risk factors in individuals, including people with diabetes. Second, we included randomized controlled trials that were primarily high quality to give the greatest protection against bias; we used available intention-to-treat data that tended to provide more conservative pooled estimates [75]; and the GRADE approach assessed the overall quality of the evidence.

GRADE is to assess the confidence we have in observed results. A high rating means the point estimate is reflective of the association between exposure and outcome, moderate means the point estimate is probably reflective of the association between exposure and outcome, low means the point estimate may be reflective of the association between exposure and outcome, and very low means we do not know whether the point estimate is at all reflective of the association between exposure and outcome. It is important to note that the evidence coming from observational studies starts with a low certainty of evidence and may be further downgraded, particularly due to indirectness and/or imprecision.

A limitation of our systematic reviews and meta-analyses was evidence of serious imprecision in the pooled estimates across several outcomes. In some cases, clinically important harm could not be ruled out because the 95%CIs were too wide. Some variables lost significance due to instability in the significance of the pooled effect estimates from the removal of single trials during sensitivity analyses. There was also complication in the pooled estimates for a few outcomes due to serious indirectness, as well as evidence of inconsistency in several variables. The inclusion of the meta-analysis on the cardiometabolic benefits of the Nordic diet is another limitation, as it has not been graded for certainty of evidence. Other limitations include analyzing dietary patterns due to reliability in self-reported data, adherence, and variations in food processing. There are also potential sources of confounding, such as collinearity effects from factors related to diet and lifestyle.

### 4.5. Implications

Our review looks at different dietary patterns using studies that include people with diabetes and shows that diet plays a key role in preventing cardiovascular disease. It is important to note that while dietary patterns can affect one’s risk and management of disease, diet often fits within a lifestyle that ultimately contributes the greatest impact on health outcomes.

The number one cause of mortality in the world is cardiovascular disease, for which diabetes is a risk factor. The results of this series of meta-analyses show that the Mediterranean, DASH, and vegetarian dietary patterns play a role in preventing the incidence of and mortality from cardiovascular disease and several other outcomes. Furthermore, the DASH, Portfolio, Nordic, vegetarian and liquid meal replacement dietary patterns improve some cardiometabolic risk factors.

The Portfolio and liquid meal replacement dietary patterns were associated with reductions in cardiometabolic risk factors, but longer-term studies are needed to confirm the safety and clinical benefits in terms of cardiovascular disease prevention.

There is a need to evaluate the effects of the DASH and vegetarian diets on cardiovascular disease prevention, specifically in those with diabetes in which adhering to these healthy dietary patterns would be of great interest. All of these patterns encourage the intakes of whole grains and legumes, at least 4–5 servings of fruits and vegetables per day, and limited saturated fat intake.

Those targets are not met by Westernized dietary patterns. In fact, only 10% of Americans eat the recommended amount of fruits or vegetables [76]. According to the European Health Interview Survey Eurostat 2016, only 14.1% of the adults living in the European Union consume five portions of fruits and vegetables per day, and approximately 33% of 35 years old and above have intakes of saturated fat of ≥15% of their energy intake [77]. That saturated fat intake is rather high relative to the current recommendation of 5–6% by the American Heart Association [78]. These data suggest that there is an opportunity for individuals to experience cardiometabolic benefits by adopting a dietary pattern that includes fruits, vegetables, whole grains and legumes.

## 5. Conclusions

In conclusion, the present findings suggest that the Mediterranean, DASH, Portfolio, Nordic, and the vegetarian dietary patterns, may have positive effects on the risk of various cardiovascular disease outcomes. Additionally, these dietary patterns along with liquid meal replacements may improve several cardiometabolic risk factors. Further research will improve our certainty in the estimates for all dietary patterns. Additionally, long-term randomized trials are needed to assess the effect of the Portfolio and liquid meal replacement dietary patterns on hard cardiometabolic outcomes.

## Figures and Tables

**Figure 1 nutrients-11-02209-f001:**
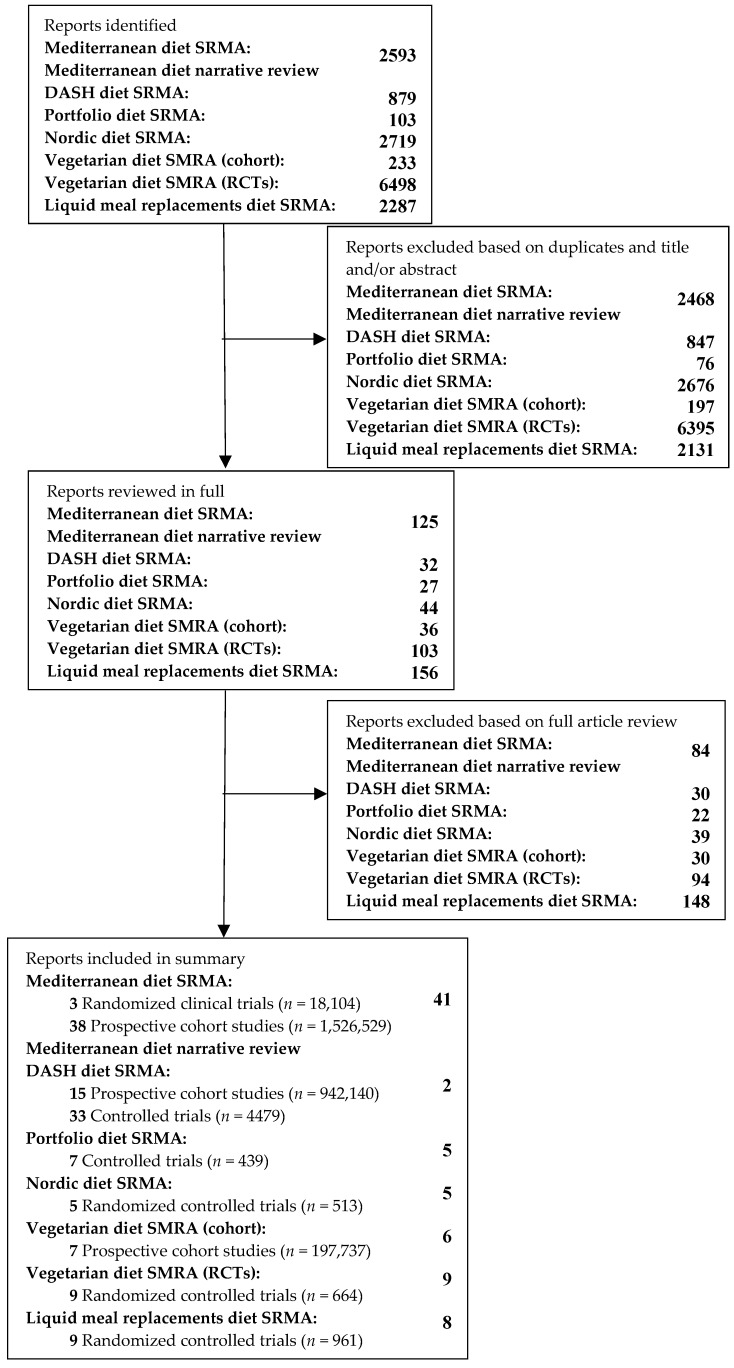
The study selection flow diagram. SRMA: Systematic review and meta-analysis. RCT: Randomized controlled trial. DASH: Dietary Approaches to Stop Hypertension.

**Figure 2 nutrients-11-02209-f002:**
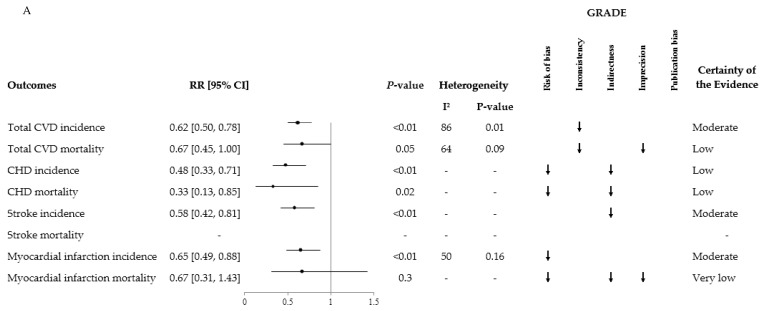
Summary and GRADE of the pooled effect estimates of prospective cohort studies, randomized controlled trials, and systematic reviews and meta-analyses assessing the association between dietary patterns and the incidence and mortality of major cardiovascular events. RR: Risk ratio. CI: confidence interval. CVD: cardiovascular disease. CHD: coronary heart disease. (**A**) Mediterranean Diet Randomized Clinical Trials. Reprinted from Critical Reviews in Food Science and Nutrition, Nerea Becerra-Tomás, Sonia Blanco Mejía, Effie Viguiliouk, Tauseef Khan, Cyril W.C. Kendall, Hana Kahleová, Dario Rahelić, John L. Sievenpiper, and Jordi Salas-Salvadó, Mediterranean Diet, Cardiovascular Disease and Mortality in Diabetes: A Systematic Review and Meta-Analysis of Prospective Cohort Studies and Randomized Clinical Trials, 2019 Taylor & Francis Ltd., https://doi.org/10.1080/10408398.2019.1565281 reprinted by permission of the publisher (Taylor & Francis Ltd., http://www.tandfonline.com). (**B**) Mediterranean Diet Prospective Cohorts. Reprinted from Critical Reviews in Food Science and Nutrition, Nerea Becerra-Tomás, Sonia Blanco Mejía, Effie Viguiliouk, Tauseef Khan, Cyril W.C. Kendall, Hana Kahleová, Dario Rahelić, John L. Sievenpiper, and Jordi Salas-Salvadó, Mediterranean Diet, Cardiovascular Disease and Mortality in Diabetes: A Systematic Review and Meta-Analysis of Prospective Cohort Studies and Randomized Clinical Trials, 2019 Taylor & Francis Ltd., https://doi.org/10.1080/10408398.2019.1565281 reprinted by permission of the publisher (Taylor & Francis Ltd., http://www.tandfonline.com). (**C**) Dietary Approaches to Stop Hypertension (DASH) Diet Prospective Cohorts. (**D**) Vegetarian Diet Prospective Cohorts.

**Figure 3 nutrients-11-02209-f003:**
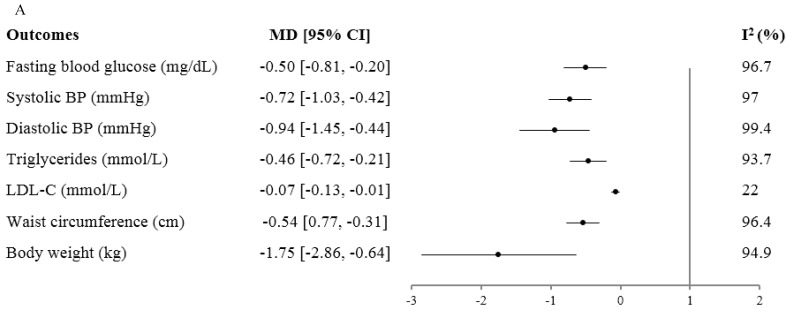
Summary and GRADE of the pooled effect estimates of prospective cohort studies, randomized controlled trials, and systematic reviews and meta-analyses assessing the association between dietary patterns and cardiometabolic risk factors. MD: mean difference. CI: confidence interval. HbA1c: hemoglobin A1c. LDL-C: low-density lipoprotein cholesterol. HDL-C: high-density lipoprotein cholesterol. Non-HDL-C: non-high-density lipoprotein cholesterol. BMI: body mass index. BP: blood pressure. ApoB: apolipoprotein B. CRP: c-reactive protein. CHD: coronary heart disease. (**A**) Mediterranean Diet Prospective Cohorts and Randomized Clinical Trials. Reprinted from Progress in Cardiovascular Diseases, Volume 61, Jordi Salad-Salvadó, Nerea Becerra-Tomás, Jesús Francisco García-Gavilán, Mònica Bulló, Laura Barrubés, Mediterranean Diet and Cardiovascular Disease Prevention: What Do We Know?, 62–67., https://doi.org/10.1016/j.pcad.2018.04.006. Copyright 2018, with permission from Elsevier. (**B**) DASH Diet Controlled Trials. (**C**) Portfolio Diet Controlled Trials. Reprinted from Progress in Cardiovascular Diseases, 61(1), Laura Chiavaroli, Stephanie K. Nishi, Tauseef A. Khan, Catherine R. Braunstein, Andrea J. Glenn, Sonia Blanco Mejia, Dario Rahelic, Hana Kahleova, Jordi Salas-Salvado, David J.A Jenkins, Cyril W.C Kendall, Johns L. Sievenpiper, Portfolio Dietary Pattern and Cardiovascular Disease: A Systematic Review and Meta-Analysis of Controlled Trials, 45–53., https://doi.org/10.1016/j.pcad.2018.05.004, Copyright (2018), with permission from Elsevier. (**D**) Liquid Meal Replacement Diet Randomized Controlled Trails. Reprinted with permission from Jarvis. C Noronha, Stephanie K. Nishi, Catherine R. Braunstein, Tauseef A. Khan, Sonia Blanco Mejia, Cyril W.C. Kendall, Hana Kahleová, Dario Rahelić, Jordi Salas-Salvadó, Lawrence A. Leiter and John L. Sievenpiper: The Effect of Liquid Meal Replacements on Cardiometabolic Risk Factors in Overweight/Obese Individuals With Type 2 Diabetes: A systematic Review and Meta-analysis of Randomized Controlled Trials, Diabetes Care 2019 May; 42(5): 767–776: https://doi.org/10.2337/dc18-2270. Copyright 2019 by the American Diabetes Association. (**E**) Vegetarian Diet Randomized Controlled Trials. Reprinted from Clinical Nutrition, 38(3), Effie Viguiliouk, Cyril WC. Kendall, Hana Kahleová, Dario Rahelić, Jordi Salas-Salvadó, Vivian L. Choo, Sonia Blanco Mejia, Sarah E. Stewart, Lawrence A. Leiter, David JA. Jenkins, John L. Sievenpiper, Effect of Vegetarian Dietary Patterns on Cardiometabolic Risk Factors in Diabetes: A Systematic Review and Meta-Analysis of Randomized Controlled Trials, 1133–1145., https://doi.org/10.1016/j.clnu.2018.05.032, Copyright (2018), with permission from Elsevier.

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
