# Peer review of "Dietary Patterns and Cardiometabolic Outcomes in Diabetes: A Summary of Systematic Reviews and Meta-Analyses"

_nutrients, 2019, doi:10.3390/nu11092209_

Round 1

Reviewer 1 Report

Dietary patterns and cardiometabolic outcomes in 2 diabetes: A summary of systematic reviews and meta-3 analyses.

In the current study, the authors summarized the evidence from different systemic reviews and mete analyses using the (GRADE) approach of different dietary patterns and patient-important cardiometabolic outcomes to compare the effect of Mediterranean, DASH, Portfolio, Nordic, liquid meal replacement, and vegetarian dietary patterns in cardiometabolic outcomes in diabetic patients. The present study suggests that the Mediterranean, DASH, Portfolio, Nordic, liquid meal replacement and vegetarian dietary patterns have cardiometabolic advantages in populations inclusive of diabetes.

The manuscript is clearly written. However, there are some issue that need to be addressed by the authors.

Please find below specific comments.

The GRADE approach is a kind of subjective tool. What were the measures that the authors took to account for this and reduce variability when they assessed the certainty of evidence? Authors should mention clearly in the methods part how many subjects were included in their meta-analysis and how many of these were diabetic and how severe was the diabetes. In addition, where were the studies they included in the manuscript conducted? Newcastle-Ottawa Scale is mostly for non-randomized clinical trials. How did the author apply this to the randomized clinical trials they included in the current study? For some of the dietary patterns, the meta-analysis for the randomized clinical trials is not included in the text. However, the analysis is shown in the figures which makes it confusing for the reader. The authors should comment on the reason for that and make it more clear which analysis is included. For figures 2A, 2C and 2 E, the certainty of the evidence is not included. The figures should be consistent.  

Author Response

Reviewer 1

In the current study, the authors summarized the evidence from different systemic reviews and mete analyses using the (GRADE) approach of different dietary patterns and patient-important cardiometabolic outcomes to compare the effect of Mediterranean, DASH, Portfolio, Nordic, liquid meal replacement, and vegetarian dietary patterns in cardiometabolic outcomes in diabetic patients. The present study suggests that the Mediterranean, DASH, Portfolio, Nordic, liquid meal replacement and vegetarian dietary patterns have cardiometabolic advantages in populations inclusive of diabetes.

The manuscript is clearly written. However, there are some issue that need to be addressed by the authors. Please find below specific comments.

The GRADE approach is a kind of subjective tool. What were the measures that the authors took to account for this and reduce variability when they assessed the certainty of evidence?

Thank you for raising this important point. We have added more detail on the GRADE process in the Methods section.

Authors should mention clearly in the methods part how many subjects were included in their meta-analysis and how many of these were diabetic and how severe was the diabetes. In addition, where were the studies they included in the manuscript conducted?

Than you for your comment. This information has been added.

Newcastle-Ottawa Scale is mostly for non-randomized clinical trials. How did the author apply this to the randomized clinical trials they included in the current study?

Thank you for raising this question. The risk of bias was assessed by the Cochrane Risk of Bias tool in randomized controlled trials and Newcastle-Ottawa Scale was used for observational studies. This has been explained in the Methods section.

For some of the dietary patterns, the meta-analysis for the randomized clinical trials is not included in the text. However, the analysis is shown in the figures which makes it confusing for the reader. The authors should comment on the reason for that and make it more clear which analysis is included.

We apologize for the confusion. We have made it more clear which analyses are included and all of our data is now reflected in both text and figure form.

For figures 2A, 2C and 2 E, the certainty of the evidence is not included. The figures should be consistent. 

Thank you for pointing this out. We have added the information in Figures 2C and 2E. Unfortunately, we are not able to add the GRADE assessment to Figure 2A, because the GRADE system was not used in this study and the information is not available. However, we believe that this study still represents a valuable contribution to the evidence and would like to keep it in the manuscript in its present form.

Reviewer 2 Report

Ref. No.:   nutrients-572760

Kahleova et al conducted systemic review and Meta-analysis on Dietary patterns and cardiometabolic outcomes in diabetes and authors concluded low to moderate certainty for most dietary patterns investigated. Overall, the manuscript is poorly prepared and needs to be largely reworks especially in the methodology section. The definition of each dietary pattern and disease outcome in each selected paper should be clearly described in the baseline table. Authors should explain how the systemic review and Meta-analysis was conducted and provide information such as key word search, flow chart for inclusion and exclusion criteria, baseline table and major finding of the selected papers. The lack of such information makes the current finding difficult to interpret and hence no conclusion can be drawn based on the current version of the manuscript.

Definition of dietary patterns: How many paper was selected for each dietary pattern? Did each selected paper for specific dietary pattern used the same dietary components (e.g. same food item and amount)? Did authors include “modified dietary pattern (e.g. modified according to the country like it tends to occurs in Mediterranean diet)”? In reality, dietary patterns tend to differ between studies and this may, due in part, to the methods used (e.g. PCA or RRR), the food groups that were groups by the authors and the amount of food consumed by the study population. Authors only cited the reference for each dietary pattern (page 3, lines 110-116), however, all the key methods used in this analysis should be provided in the method sections in order to help reader understand how the dietary pattern was defined and how the meta-analysis was conducted. Flow chart of paper selection is missing. How paper was excluded and selected? How many were RCT and cohort studies? What are the key words searched for each disease and dietary pattern?... How did authors score the GRADE system for each selected article? For example, some figures did not have grade information (e.g. Fig 2A, 2c) and some of them only had one score (Fig 1c) How did authors define the overall certainty of the evidence of “moderate”, “low”, “very low”? please describe it in the method. Statistics: Methods for the estimation of the effect size (random or fixed effect?), Tau and I square should be reported

Results:

Quality assessment result of Newcastle-Ottawa scale should be included in the manuscript as supplemented table Baseline characters of study populations and study results were missing. Questions such as how many papers were selected for the analysis for each dietary pattern, study numbers, study duration and study outcome were not reported. Figures are inconsistently prepared. For example, the total n number and selected papers should be included in each figs. Fig 2 A and C had no information of information of “certainty of the evidence” but others had. Nordic dietary patterns: this study did NOT provide any data on Nordic dietary pattern, so, authors should remove “Nordic dietary pattern” from this manuscript including aim, methods and results sections.

Author Response

Thank you for your helpful comments. Please find our response attached.

Reviewer 3 Report

The authors have produced a recent series of systematic reviews considering dietary patterns in populations of, or those inclusive of, adults with diabetes. The current paper is a necessary synthesis of these reviews so to inform dietary guidelines for diabetes management. The authors have done an excellent job in their introduction, methods, and results. As such, I have very few comments, all minor and primarily of a clerical nature regarding these components of the manuscript.

My only comments of substance for these parts of the manuscript are: (1) to ask if the authors considered meta regression analyses to be able to comment on differences in outcomes between the dietary patterns. This would introduce a new element to this summary paper, although I recognize this may have been considered, tested, and decided against based on the amount and reliability of the data that are available. And (2) should the comparison diets be discussed and considered in relation to the findings from the controlled trials?

Minor comments on content before the discussion:

On line 87 would it not be more correct to state that ‘each review protocol was registered at ClinicalTrials.gov’, with the word study ambiguous.

The figures presented in this manuscript require standardization, although for some issues it may just be how they look when loaded into the journal’s submission portal. Figure 1C shows additional criteria not found in the other figures that would lead to an upgrade or downgrade of evidence. Could they be removed? Figure 2A and 2C are not graded, the forest plots of figure 2C and 2D do not have Y axis, and many of the point estimates of 2E do not have confidence intervals. Furthermore, for readers not familiar with GRADE I wonder if it would be helpful to use an up or down arrow instead of a filled square when indicating a change in evidence rating due to the risk of bias, inconsistency, indirectness, imprecision, or dose response.

I think the discussion could be modified to better inform the reader.

4.1 There is useful content on lines 384-399 that may be better placed within this summary of main findings.

4.2 does not currently relay much useful information, findings from the author’s recent series of systematic reviews are not compared with the results of previous reviews that are now superseded. If there were changes in the direction of findings between these reviews and those of the past, that would be important to know. If there are not changes, the current authors present the most up-to-date information available utilizing more data than previous works. This is also important to know. This section may also be an appropriate place to contrast the dietary pattern data with recent food group data from authors such as Schwingshackl, Aune, and Reynolds. That patterns are based on food groups, and that there is consistency between food groups and patterns in the literature is an important addition to this discussion.

4.3 could be better tailored to consider the mechanisms supporting why adhering with these dietary patterns would produce the observed effects. Line 332 -333 appears to be an obvious place to refer to a recent systematic review of dietary fibre in human health. Greater satiety reducing dietary intake is a further a mechanism that could be discussed. Lower intakes of saturated fats is not a mechanism in itself, but its role in reducing atherosclerosis would be. The specific role of plant protein beyond total protein (controlling for other dietary components such as saturated fat and sodium), and low glycemic index foods (controlling for other dietary components such as dietary fibre) in a wide range of cardiometabolic risk factors would need to be much better supported by references to relevant literature to be included in this section. Mechanisms currently attributed to only the Portfolio diet could be discussed in the context of those common food groups emphasized by all patterns. Given that this article is a synthesis of the recent systematic reviews, a comparison of the similarities between patterns in terms of the foods promoted or discouraged is necessary.

4.4 Limitations should relate more broadly to your processes and findings, and not be specific to GRADE. Current limit 1 and 4 seem appropriate with some further specificity on what outcomes had confidence intervals that included harm. If the authors identified limitations within each systematic review, such as participant adherence in RCTs or potential variability in the health outcomes from patterns due to differences in food processing, this may also be identified as limitations and discussed to better inform the reader.

4.5 text on lines 384-399 does not relate to implications. Much of the rest of the text is very good, however please consider an acknowledgement and additional discussion that while dietary patterns are clearly important, they fit within a lifestyle that ultimately impact diabetes risk and management.

General comments on discussion wording:

Line 276 suggest rephrase

Line 289 suggest increase clarity of phrasing (inclusion into what?)

Line 290-291 suggest rephrase

Only a small suggestion to consider in the conclusion. Is the phrase ‘more research is needed to improve of certainty in the estimates’ relevant for all your patterns and all your findings, or if this may be removed so that the reader may focus on your targeted comment for RCTs on the Portfolio and liquid meal replacement diets?

Author Response

Thank you for your comments. Please find our response attached.

Round 2

Reviewer 2 Report

The revised manuscript has substantially improve it's clarity in the methodology. Just  one question left for authors: How do authors explain when the RR is significant but the GRADE suggests low or very low certainty of the evidence? For example, Fig 2B showed siginificant effects of Mediterranean diet prospective cohorts on CHD mortality/stroke incidence and mortality/myocardial infarction incidence and mortality but certainty of the evidence is low or very low. Please discuss it in the discussion.

Author Response

Thank you for your insightful comments and for your commitment to help us improve the manuscript. We appreciate your wonderful help.

Reviewer 2

The revised manuscript has substantially improve it's clarity in the methodology. Just  one question left for authors: How do authors explain when the RR is significant but the GRADE suggests low or very low certainty of the evidence? For example, Fig 2B showed siginificant effects of Mediterranean diet prospective cohorts on CHD mortality/stroke incidence and mortality/myocardial infarction incidence and mortality but certainty of the evidence is low or very low. Please discuss it in the discussion.

Thank you for this great question. GRADE is to assess the confidence we have in observed results. A high rating means the point estimate IS reflective of the association between exposure and outcome, moderate means the point estimate IS PROBABLY reflective of the association between exposure and outcome, low means the point estimate MAY BE reflective of the association between exposure and outcome, and very low means WE DON’T KNOW if the point estimate is at all reflective of the association between exposure and outcome. It is important to note that the evidence coming from observational studies starts with a low certainty of evidence and may be further downgraded, particularly due to indirectness and/or imprecision. We have added this clarification in the Discussion section.